# Strategies to Augment Natural Killer (NK) Cell Activity against Solid Tumors

**DOI:** 10.3390/cancers11071040

**Published:** 2019-07-23

**Authors:** Ziqing Chen, Ying Yang, Lisa L. Liu, Andreas Lundqvist

**Affiliations:** Department of Oncology-Pathology, Karolinska Institutet, S-17164 Stockholm, Sweden

**Keywords:** natural killer cells, immune checkpoint, tumor microenvironment

## Abstract

The immune system plays a crucial role to prevent local growth and dissemination of cancer. Therapies based on activating the immune system can result in beneficial responses in patients with metastatic disease. Treatment with antibodies targeting the immunological checkpoint axis PD-1 / PD-L1 can result in the induction of anti-tumor T cell activation leading to meaningful long-lasting clinical responses. Still, many patients acquire resistance or develop dose-limiting toxicities to these therapies. Analysis of tumors from patients who progress on anti-PD-1 treatment reveal defective interferon-signaling and antigen presentation, resulting in immune escape from T cell-mediated attack. Natural killer (NK) cells are innate lymphocytes that can kill tumor cells without prior sensitization to antigens and can be activated to kill tumor cells that have an impaired antigen processing and presentation machinery. Thus, NK cells may serve as useful effectors against tumor cells that have become resistant to classical immune checkpoint therapy. Various approaches to activate NK cells are being increasingly explored in clinical trials against cancer. While clinical benefit has been demonstrated in patients with acute myeloid leukemia receiving haploidentical NK cells, responses in patients with solid tumors are so far less encouraging. Several hurdles need to be overcome to provide meaningful clinical responses in patients with solid tumors. Here we review the recent developments to augment NK cell responses against solid tumors with regards to cytokine therapy, adoptive infusion of NK cells, NK cell engagers, and NK cell immune checkpoints.

## 1. Introduction 

Immunotherapy has quickly become one of the corner stones in modern cancer treatment. Current approaches in immunotherapy include the administration of cytokines, antibodies and more advanced manipulations involving *in vitro* expansion of T cell and Natural killer (NK) cell subsets [1,2,3,4]. 

NK cells were discovered in the mid 70’s based on their “natural “ capacity to kill tumor cells without prior sensitization [5,6]. In contrast to T cells, NK cells sense the absence of self Major Histocompatibility Complex (MHC) class I molecules through stochastically expressed inhibitory receptors. This suggests that NK cells may be particularly effective when transferred across HLA barriers [7,8]. In addition to antibody-independent cytotoxicity, the expression of CD16 on a majority of NK cells renders them strong mediators of antibody dependent cellular cytotoxicity (ADCC). Taking advantage of this, various mAbs have been developed and have now become the standard of care in various hematological and solid cancers, including rituximab, cetuximab and trastuzumab. Other routes by which NK cells can kill targets are the death receptor pathways Tumor necrosis factor (TNF)-related apoptosis-inducing ligand (TRAIL)/TRAIL-R and Fas/FasL. Instead of triggering the release of cytotoxic granules, death receptor pathways prompt apoptosis via caspase activation in target cells.

Although NK cell therapy has been successful in targeting hematological malignancies, the outcome of adoptive NK cell infusion into patients with solid tumors has been rather disappointing. One of the major challenges with NK cell-based therapies against solid tumors includes trafficking of NK cells to the tumor location and infiltration into the tumor. Several studies have shown that there is a correlation between the presence of NK cells at the tumor site and tumor progression [9,10,11]. Furthermore, the presence of inhibitory signals within the tumor microenvironment and altered immunogenicity of tumor cells also contributes to the poor infiltration and activation of NK cells at the tumor site [12]. 

Increasing interest in NK cells over the past years has resulted in several ongoing clinical trials beginning to systematically address the potential role of NK cells in clinical settings. Intense research effort is now made to enhance NK cell function to target tumors. In this review we will discuss the recent developments in augmenting NK cell responses against solid tumors. 

## 2. Cytokines

Growth factors that belong to the common γ-chain cytokines including interleukin-2 (IL-2), IL-4, IL-7, IL-9, IL-15 and IL-21, play key roles in the development and homeostasis of T and NK cells [13]. IL-2 activates NK cells via binding to the heterotrimeric IL-2 receptor that consists of the IL-2 receptor subunit alpha (CD25) and beta (CD122), and the common gamma chain (CD132). Patients undergoing treatment with adoptive NK cell therapy are often given IL-2 to sustain the *in vivo* expansion of infused NK cells [13,14,15] (Table 1
^1,2,3^). However, recombinant IL-2 has a limited half-life and is associated with dose-limiting adverse events such as arrhythmias, heart failure and capillary leak syndrome that lead to life-threatening toxicities in patients. While the administration of low-dose IL-2 show a lower toxicity profile, little clinical benefit of IL-2 therapy was detected in matched-pairs analysis [14]. Weekly administration of IL-2 together with interferon-α can lead to exhaustion of NK cells, which may explain the low efficacy of IL-2 as a monotherapy [16]. 

Another potential explanation for the lack of efficacy of IL-2 treatment is the expansion of regulatory T cell (Treg) that will interfere with the generation of anti-tumor responses. Upon activation with IL-2, Treg produce high amounts of Transforming Growth Factor (TGF)β to suppress T and NK cell responses [15,17]. Treg can also interfere with NK cell responses by sequestering IL-2 via the high affinity IL-2 receptor complex. 

Several efforts have been made to engineer or modify IL-2 to improve its therapeutic potential. Levin et al. engineered an IL-2 “superkine” (or super-2) where the functional requirement of IL-2 for CD25 was eliminated but with increased binding affinity for IL-2Rβ. Compared with native IL-2, IL-2 superkine induce activation of T and NK cells resulting in improved anti-tumor responses *in vivo* with limited expansion of Treg [18]. Recently, Silva and colleagues designed a mimic of IL-2 that bind both CD122 and CD132 without binding to CD25 or the IL-15Rα (CD215). This compound, Neo-2/15, showed enhanced activation of murine and human T and NK cells and superior therapeutic activity compared with IL-2 in murine models of melanoma and colorectal cancer [19]. A third approach has been developed by Nektar Therapeutics. NKTR-214 is a clinical-stage biologic that comprises the IL2 protein bound by multiple releasable polyethylene glycol (PEG) chains. The PEG chains are located at the region of IL2 that contacts the CD25 of the heterotrimeric IL2 receptor complex, thereby reducing its ability to bind and activate the heterotrimer. As a consequence, treatment with NKTR-214 resulted in preferential activation of T and NK cells with limited activation of Treg within the tumor microenvironment and provided anti-tumor efficacy across several syngeneic models [20]. Administration of NKTR-214 has recently shown to be well tolerated with clinical activity including tumor shrinkage in heavily pre-treated patients [21].

Another type I cytokine that can activate and expand NK cells that show promising therapeutic potential is IL-15. The IL-15 receptor complex consists of CD122, CD132, and CD215 [22,23]. Despite sharing the common gamma receptor and the beta subunit and the same signaling subunits, the gene expression signature in lymphocytes differ between IL-15 and IL-2 [24]. Recently, IL-15 treated NK cells were shown to maintain anti-tumor activities in the context of an immunosuppressive microenvironment compared with IL-2 treated NK cells [25,26,27]. These findings argue that treatment with IL-15 may provide a better anti-tumor effect compared with IL-2. However, a recent study showed that continuous IL-15 signaling impairs NK cell anti-tumor activity through a metabolic defect [28]. This finding highlighted the need for a rational design for dosing and administration schedule of IL-15. IL-15 is currently being evaluated in numerous clinical trials, either alone or in combination with adoptive NK cell treatment in patients with solid and hematological cancers (Table 1
^4,5^). Recently, Conlon et al. reported results from a first-in-human clinical trial with rhIL-15 administered as a 10-day continuous intravenous infusion to patients with cancers. At 2 µg/kg/day, circulating CD8+ T cells and NK cells increased by 5.8- and 38-fold respectively. Among the 27 included patients, 17 had stable disease as the best response [29]. To increase the efficacy of IL-15 a superagonist IL-15-IL15Rα-Sushi-Fc fusion complex (ALT803) was developed. This compound shows greater biological activity than native IL-15 with increased potency to stimulate NK cell anti-tumor activity [30]. In a recent first-in-human phase 1 clinical trial, ALT-803 was administered in patients who relapsed after allogeneic hematopoietic stem cell transplantation. Responses were observed in 19% of evaluable patients, including one complete remission lasting seven months. While subcutaneous administration of ALT-803 did not result in any dose-limiting toxicities, intravenous administration included constitutional symptoms temporally related to increased serum IL-6 and Interferon (IFN)γ which may limit dosing and therefore its therapeutic efficacy [31].

Efforts to target cytokines to the tumor microenvironment to potentially reduce systemic adverse events are being pursued. Immunocytokines are proteins of tumor-specific antibodies fused to a cytokine that can deliver cytokines into the tumor microenvironment. Immunocytokines, particularly IL-2, have shown promising anti-tumor effects in several murine tumor models and a few have entered clinical trials [32]. Vincent et al. showed that an immunocytokine consisting of the IL-15Rα linked to IL-15 with a tumor specificity for the GD2 ganglioside had strong antitumor activity in syngeneic cancer models [33]. The immunocytokine N-809, which comprises of ALT-803 fused to two single-chain anti-PD-L1 domains was recently described. N-809 was shown to enhance the cytotoxic potential of both CD8+ T cells and NK cells via ADCC. In the MC38-CEA murine colon carcinoma model, 60% of mice treated with N-809 underwent complete tumor rejection [34]. Similarly, Wrangle et al. recently showed patients with non-small cell lung cancer treated with the combination of ALT803 and nivolumab showed a remarkable expansion and proliferation rate of NK cells. Objective response was observed in 6 or 21 patients (29%). Nine patients (43%) had a decrease in target lesion size, and disease control was observed in 16 patients (76%). Median progression-free survival and overall survival was 9.4 months and 17.4 months respectively [35]. 

Improvement in cytokine technology has provided evidence of clinical efficacy in patients with cancer in recent years. Other members of the type I cytokine family are being developed for clinical use. For example, IL-21 has shown to mediate the reversal of NK cell exhaustion [36]. Furthermore, combinations of different cytokines can potentiate NK cell activity compared with the single use of individual cytokines. The combination of IL-12, IL-15, and IL-18 can induce human memory-like NK cells with elevated and prolonged IFNγ production and improved targeting of leukemia cells in vivo [37,38,39]. 

## 3. Adoptive Cell Therapy

The adoptive transfer of NK cells has shown impressive clinical results in patients with AML and myelodysplastic syndrome (MDS) in Killer-cell immunoglobulin-like receptor (KIR)-ligand mismatched settings [40,41,42,43]. However, adoptive infusion of NK cells in patients with solid tumors has so far shown less clinical efficacy. In the early 1980s, Rosenberg and collaborators explored the therapeutic potential of autologous lymphokine activated killer (LAK) cells in combination with high-dose IL-2 in patients with advanced metastatic renal cancer and melanoma [44]. Objective clinical regression was observed in 11 out of 25 patients and complete tumor regression occurred in one patient with metastatic melanoma. In a randomized phase III clinical trial in patients with advanced renal cell carcinoma showed a median survival of 13 months but no difference between patients receiving IL-2 alone or in combination with LAK cells was observed [45]. In these studies, LAK cells were generated from unselected mononuclear cells and not purified NK cells. In a clinical trial where selected *ex vivo* expanded NK cells were infused, no clinical response despite the persistence of infused NK cells was observed [46]. These persistent NK cells were able to mediate ADCC responses without cytokine reactivation *in vitro* suggesting that combining adoptive NK cell transfer with monoclonal antibody administration deserves evaluation. Ishikawa et al. explored NK cell transfer in combination with trastuzumab or cetuximab in advanced gastric or colorectal cancer patients. Four out of six patients presented with stable disease and increased IFNγ levels and reduced regulatory T cells frequencies in peripheral blood suggesting an enhanced NK cell activity [47]. However, Talleur et al. showed limited clinical benefit of infusion of haploidentical NK cells when combined with anti-GD2 treatment in pediatric neuroblastoma patients that underwent autologous hematopoietic cell transplantation [48]. Iliopoulou et al. explored infusion of allogeneic donor NK cells in 15 patients with non-small cell lung cancer. Although local or systemic adverse events were absent, only two patients presented with a partial response and six patients with a disease stabilization [49]. Geller et al. evaluated the infusion of allogeneic NK cells following conditioning with fludarabine, cyclophosphamide, and total body irradiation in patients with ovarian and breast cancer. The authors reported a transient donor chimerism and concluded that strategies to augment *in vivo* NK cell persistence and expansion should be explored [50]. 

In order to improve persistence within a solid tumor microenvironment, adoptively transferred NK cells need to overcome an array of immunosuppressive factors. A recent phase I trial showed stable disease in 8 of 17 patients with lymphoma and solid tumors following three infusions of allogeneic NK cells. Immune monitoring revealed that NKG2D positive T cells significantly increased along with a decreased frequency of Treg, myeloid-derived suppressor cells (MDSC) and TGFβ production, suggesting that the infused NK cell could boost T cell activation (Table 1
^6^) [51]. Factors regulating NK cell activity include IL-10, TGFβ, IL-6, prostaglandin-E2 (PGE2), indoleamine-2,3-dioxygenase (IDO), and adenosine (reviewed in [52,53]). These factors can be produced by the tumor cells themselves or Treg, MDSCs, cancer-associated fibroblasts (CAFs), dendritic cells, and tumor-associated macrophages. We found that tumor cells produce PGE2 to convert monocytes into MDSCs and these MDSCs potently inhibited NK cell activity through the production of TGFβ. Silencing COX-2 reduced the accumulation of peripheral blood MDSCs, resulting in concomitant improved *in vivo* clearance of tumor cells [54]. Interestingly, Sarhan and colleagues showed that human cytomegalovirus (CMV)-induced adaptive NK cells were able to resist suppression by MDSCs and Tregs [55,56]. 

Another reason for the poor clinical outcome of adoptive infusion of NK cells in patients with solid tumors is the insufficient migration of the infused cells toward the tumor. Upon *ex vivo* expansion and activation of NK cells, the chemokine receptor repertoire changes. For example, NK cells upregulate the expression of CXCR3 upon activation. This facilitates the migration towards the IFNγ-inducible chemokines CXCL9, 10, and 11. Kim et al. recently showed that CXCR3-deficient NK cells failed to migrate towards CXCL10 positive B16 melanoma tumors *in vivo* [57]. Similarly, we found that activated CXCR3 positive human NK cells selectively migrated towards CXCL10 positive human melanoma tumors in an *in vivo* xenograft model [58]. However, CXCL10 is only produced in abundant levels in inflamed tumors. Instead, solid tumors often produce high levels of IL-8 and growth-related oncogene –alpha, beta, and gamma. These chemokines recruit cells expressing the CXCR1 and CXCR2 receptors [59]. While peripheral blood NK cells express the CXCR2 receptor, it is rapidly lost upon *ex vivo* activation. We recently showed that genetic engineering of activated human NK cells to express the CXCR2 receptor facilitated the migration of these cells toward IL-8 producing tumor cells [60]. 

Another aspect that needs to be considered to improve the clinical outcome of adoptive infusion of NK cells is to enhance the ability to directly target tumor cells. This can be achieved through genetic engineering by chimeric antigen receptors (CAR) or T cell receptors. CAR-NK cells targeting B cell malignancies has demonstrated impressive efficacy *in vivo* [61,62,63]. Liu et al. showed that cord blood NK cells engineered to express IL-15 and a CD19-CAR showed a marked increase in survival in a xenograft lymphoma model [64]. To date, two clinical trials with genetically modified NK cells in patients with solid tumors are enrolling patients (Table 1
^7,8^). Recently, two reports showed that NK-92 cells can be engineered to express T cell receptors (TCRs) with specificity for melanoma antigens [65,66]. Another approach to augment NK cell-mediated killing of tumor cells is to use low-dose chemotherapy. Several independent reports have shown that low-dose chemotherapy induce sensitivity to NK cell-mediated killing through upregulation of ligands to the activating NK receptor NKG2D or receptors to the death ligand TRAIL [67,68,69,70,71]. We found that sub-apoptotic doses of the proteasome inhibitor bortezomib upregulated TRAIL-R2 on tumor cells and rendered them more sensitive to killing by NK cells. *In vivo*, combined treatment with bortezomib and infusion of NK cells resulted in significantly prolonged survival in tumor-bearing mice [72]. Based on these observations we commenced on a clinical trial where patients with metastatic cancer were treated with bortezomib combined with the infusion of autologous *ex vivo* expanded NK cells. We reported that 7/14 patients had stable diseases, including two patients who had more than a 30% decline in serum tumor markers and one patient with metastatic kidney cancer who had a minor response [73]. Similar to bortezomib, we found that the histone deacetylase inhibitor Depsipeptide also sensitized tumor cells to NK cell-mediated killing via upregulation of TRAIL receptors [68]. Other reports have found that histone deacetylase inhibitors can induce the expression of NKG2D ligands on tumor cells to render them susceptible to killing by NK cells [74,75,76,77]. In recent years, NK cells generated from induced pluripotent stem cells (iPSC-NK) or human embryonic stem cells (hESC-NK) have gained more interest. Li et al. showed that infusion of iPSC-NK directed against the mesothelin antigen significantly prolonged survival compared with infusion of peripheral blood (PB)- or iPSC-derived NK cells [78].

## 4. Natural Killer Cells Engagers

Although monoclonal antibody-based therapy continues to be improved, a significant number of patients do not benefit from antibody therapy. Monoclonal antibodies (mAbs) can trigger a variety of effector mechanisms including Fc-mediated effector functions such as ADCC by NK cells. Optimization of the antibody molecule to increase the therapeutic efficacy is a major area in current translational research. More than two decades ago, the impact of Fc glycosylation on ADCC activity was shown using CAMPATH-1H expressed in different cell lines resulting in distinct glycosylation patterns [79]. Since then, various strategies have been developed to engineer mAbs to increase their ADCC activity. Umana et al. showed that glyco-engineering of chCE7, an anti-neuroblastoma chimeric IgG1 mAb, increased NK cell mediated ADCC by 20-fold [80]. Similar results have been observed for glyco-engineered of rituximab and CD19 [81,82]. Another approach to increase ADCC is via the exchange of amino acids directly in the Fragment crystallizable gamma receptor (FcγR) binding site. Lazar et al. developed an Fc variant with enhanced FcγRII/IIIa binding affinity by amino acid substitution. This mAb showed increased ability to induce NK cell-mediated ADCC compared with native IgG1 molecules and in the context of rituximab, depleted 50 % of circulating B cells at approximately 50-fold lower dose than the non-engineered IgG1 counterpart [83]. The same or similar Fc variants were recently shown to enhance ADCC activity also in CD33 or CD133 antibodies against AML [84,85].

In the past decade, increased knowledge of NK cell receptor biology has stimulated the development of bispecific mAbs to increase specificity and facilitate a more direct interaction between immune cells and tumors while decreasing systemic toxicity. Since the introduction of bispecific mAb targeting CD16 on NK cells and CD30 on Hodgkin’s lymphoma more than two decades ago [86], bi-specific mAb constructs have been engineered to engage CD16 on NK cells and various tumor antigens including; CD19 and HLA class II for B cell malignancies [87,88], CD30 for Hodgkin’s lymphoma [89], epidermal growth factor receptor (EGFR) [90], which is overexpressed in several epithelial cancer types, HER2 for breast cancer [91], CD33 for AML [92,93] and EPCAM for carcinomas [94]. Besides engaging CD16 on NK cells, bi-specific mAbs have been designed to target other NK cell receptors including NKG2D [86,95], and NKp30 [96]. Furthermore, bispecific proteins using the NKG2D ligands MICA or ULBP2, fused together with a tumor targeting variable fragment (Fv) have shown to induce NK cell-mediated killing of target cells [97,98,99]. Von Strandmann et al. showed that the bispecific protein ULBP2-BB4 targeting NKG2D and CD138 activated NK cell antitumor activity against human multiple myeloma in vitro and in vivo [99]. 

To further increase tumor selectivity, tri-specific engagement of NK cells with dual targeting of tumor antigens has been explored. Gantke et al. showed a superior in vitro potency of a trispecific mAb targeting CD16, BCMA, and CD200 compared with bispecific mAbs targeting CD16 and BCMA or CD200 [100]. A similar approach was described by Gauthier et al., where dual engagement of the NK cell receptors NKp46 and CD16 coupled with a CD19 targeting domain resulted in a significant delayed tumor progression in vivo [101]. Another approach was developed by Vallera et al., where IL-15 as incorporated into the design of a bispecific mAb targeting CD33 and CD16, and observed prolonged in vivo persistence, activation, and survival of NK cells [102]. Interestingly, this trispecific mAb also rendered NK cells less susceptible to suppression by MDSCs [103]. 

## 5. NK Cell Immune Checkpoints

Upon infiltration into a tumor mass, NK cells undergo phenotypic changes and often show impaired function [104]. Therefore, various therapeutic approaches are being developed with the aim to restore tumor-infiltrating NK cell-mediated cytotoxicity. The signaling through inhibitory receptors prevent the activation of NK cells, and by using mAbs blocking these pathways augment the anti-tumor activity of NK cells (Figure 1).

NK cells express inhibitory receptors that bind to MHC class I molecules, which are termed Ly49 receptors in mice and Killer immunoglobulin-like receptors (KIRs) in humans. The interaction of these receptors not only mediates tolerance of NK cells to self-tissue but also plays a role in “licensing” NK cell immune response [57]. The upregulation of HLA class I expression by tumor cells and the engagement with inhibitory receptor signaling dampen the activation of NK cells. Apart from NK cells, the inhibitory KIRs are also expressed on a subset of effector and/or memory CD4+ and CD8+ T cells, so blocking KIRs might improve the antitumor activity of T cells as well [105] (Figure 1). IPH2101 is a fully human IgG4 mAb targeting KIR2DL-1, -2, -3 receptors [106]. By blocking the KIR-ligand interactions and their inhibitory signaling, IPH2101 can enhance the cytotoxicity of NK cells against tumor cells. The therapeutic potential of IPH2101 has already been shown in preclinical mouse models of AML [106], in multiple myeloma [107], and in B cell lymphoma [108]. Phase I clinical trials have shown that IPH2101 administration can block KIRs for a prolonged time and is well tolerated in the patients with AML [109] and multiple myeloma [110]. Despite that *in vitro* studies suggest IPH2101 enhances NK cell-mediated killing of KIR-ligand matched tumors without having effect on targets that are MHC class I-deficient [111], the subsequent phase II clinical trial of IPH2101 failed to show any clinical efficacy in patients with smoldering multiple myeloma [112]. The failure may be due to mAbs targeting KIRs decreasing the KIR2D+NK subset and inducing hypo-responsiveness NK cells [112,113]. Lirilumab (IPH2102, BMS-986015,LIRI), a second generation anti-KIRs mAb, has been evaluated for the safety at six dose levels in patients with hematologic malignancies and solid tumors in a phase I clinical trial [114]. Despite its limited efficacy in the early clinical trials as a single agent, anti-KIR mAb might synergize with other agents. Several studies already showed encouraging results to support a combination therapy, such as lenalidomide [107], rituximab [108], or daratumumab [115] in patients with multiple myeloma and lymphoma.

NKG2A-CD94, an important heterodimeric inhibitory receptor related to C-type lectins, is universally expressed on NK cells and some T cells. It binds to HLA-E, the non-canonical MHC class I molecule, and suppresses the NK cell activation signaling pathway [116]. The upregulation of HLA-E by hematologic malignancies or solid tumors facilitate their escape NK and T cell mediated killing [117,118]. Consistently, utilization of genetic interventions or mAbs to target the HLA-E-NKG2A pathway could be a therapeutic strategy to boost the efficacy of cancer vaccines [119]. In vitro and in vivo studies revealed that the knockdown of NKG2A in NK cells using shRNA or siRNA instigates “missing-self” responses and promotes the anti-tumor activity against HLA-E expressing cancer cell lines [120,121]. Recently Kamiya et al. developed an approach to generate NK cells lacking NKG2A expression by transduction of NKG2A protein expression blockers resulting in increased cytotoxicity towards HLA-E expressing tumor cells [122]. Monalizumab (IPH2201), a first-in class blocking mAb to NKG2A, is currently being tested in several phase I/II clinical trials for the safety and antitumor activity in various cancers, including chronic lymphocytic leukemia [123], squamous cell carcinoma of the head and neck (SCCHN), gynecologic cancers and advanced colorectal cancers (Table 1
^9,10,11,12^) [124]. In the context of advanced solid tumors, the combination monalizumab and durvalumab (anti-PD-L1mAb) is well tolerated (Table 1
^13^). A preliminary result from a phase II clinical trial investigating combined monalizumab plus cetuximab treatment in patients with SCCHN provided encouraging results with 31% objective response rate [125]. 

In addition to KIRs and NKG2A-CD94, an emerging group of inhibitory receptors that interact with nectin and nectin-like molecules includes TIGIT, CD96 (TACTILE) and CD226 (DNAM-1) [126]. Upon binding nectin and nectin-like molecules, CD226 activates NK cytotoxicity whereas TIGIT and CD96 counterbalance CD226. In tumor-bearing mice or in patients with colorectal cancer, blocking TIGIT prevented the exhaustion of NK cells and enhanced NK cell-mediate tumor immunity [126]. Blockade of TIGIT additionally promoted therapeutic effect of anti-PD1/PD-L1 mAbs [95]. Genetic depletion of CD96 show increased control by NK cells to target of metastases [127]. Likewise, another study described a role of CD96 in constraining NK cell-mediated suppression of tumor metastasis in mice. This study additionally demonstrated that CD96 mAbs also worked synergistically with chemotherapy and immune checkpoint blockades targeting PD-1 or CTLA-4 [128]. Thus, neutralizing TIGIT/CD96 alone or in combination with other agents might be a potential strategy to improve anti-tumor immunotherapy by NK cells.

Another emerging NK cell specific immune checkpoint is the cytokine-inducible SH2-containing protein (CIS). CIS is encoded by the *CISH* gene and was shown to be a negative regulator of IL-15 signaling in NK cells. Deletion of CIS in NK cells induced hypersensitiveness to IL15 and reduced experimental metastasis *in vivo* [129]. Moreover, the combination of *CISH*-depletion together with other inhibitors or immune checkpoint blockade antibodies showed improved control of tumor metastasis *in vivo* [130]. Interleukin-1 receptor 8 (IL-1R8) is a member of the IL-1 receptor (ILR) family that is involved in negative regulation of ILR and Toll-like receptor (TLR) downstream signaling pathways and orchestrate innate and adaptive immunity [131]. IL-1R8 is widely expressed on immune cells. Recently, Molgora et al. identified IL-1R8 as a negative regulator of NK cell-mediated cytotoxicity against solid tumors and indicating that IL-1R8 serves as an immune checkpoint molecule for NK cell maturation and cytotoxicity [132]. 

The immunological checkpoint molecules PD-1 and CTLA-4 can also be expressed by NK cells. A recent study identified a subpopulation of NK cells that express high levels of PD-1 in patients with ovarian carcinoma [133]. Blocking antibodies against CTLA-4 or PD-1 have displayed remarkable therapeutic effects in patients with advanced cancers [134,135,136], but little attention has been paid to NK cells in this context. *In vitro*, PD-1/PD-L1 blockade has been shown to augment NK cell-mediated tumor lysis in multiple myeloma [137]. These studies indicated that targeting PD-1/PD-L1 might also be effective for the activation of NK cells. Another potential NK cell immune checkpoint is TIM-3. The expression of TIM-3 was found to be elevated on tumor-infiltrating and peripheral blood NK cells from patients with various cancers, including gastric cancer [138,139], metastatic melanoma [140] and lung adenocarcinoma [141]. The levels of TIM-3 expression were correlated with the stage and prognosis of melanoma, and blocking TIM-3 rescued exhausted NK cells from melanoma patients [140]. LAG-3 (CD223) is emerging as a potential target for cancer immunotherapy, due to its capacity of negatively regulating T cell activation and synergizing with PD-1 to exhaust T cells [142]. Notably, LAG3 can also be expressed by NK cells. Increased LAG-3 expression was found on intratumoral NK cells in patients with renal cell carcinoma [143]. Elevated expression of LAG3 has been associated with NK cell memory and exhaustion. At present, there are several LAG3-targeting agents are under development, such as IMP321 (a soluble LAG-3 Ig), BMS-986016 (Relatlimab, anti-LAG-3 mAb) [144]. Several ongoing clinical trials are exploring the therapeutic efficacy of LAG3 and PD-1 combined treatment with various advanced cancers (Table 1
^14,15,16^) [145]. 

## 6. Concluding Remarks

The recent clinical success of targeting immune escape mechanisms like the immunological checkpoint axis PD1 – PD-L1 has transformed modern day cancer therapies. However, many patients do not respond to this therapy due to primary or acquired resistance. In order to increase the frequency of responding patients there is now significant interest in the activation of the innate immunity and in particular NK cells. In the past few years technological and methodological advancements have led to a deeper understanding of NK cell activity in solid tumors. Advancements in cytokine engineering, manufacture of cellular products for adoptive infusion, genetic engineering, and multi-specific antibodies as well as antibodies targeting immune checkpoints have led to the launch of many clinical trials to explore the potential of NK cells in the past few years. The majority of these trials show little or limited toxicity, but still not as good as clinical responses observed in haematological malignancies. To improve clinical responses in patients with solid tumor, several aspects need to be taken into account. These include but are not limited to; optimal source of NK cells, appropriate ex vivo activation to manufacture an optimal cellular product, migration of infused NK cells toward the tumor, and how to maintain the persistence of activated NK cell within an immunosuppressive solid tumor microenvironment. To achieve this, a combinatorial approach of several different therapeutic modalities is likely needed. 

## Figures and Tables

**Figure 1 cancers-11-01040-f001:**
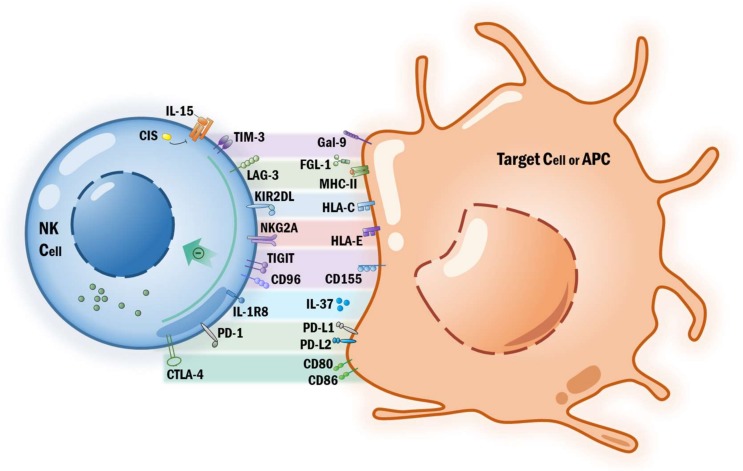
NK cell immune checkpoint.

**Table 1 cancers-11-01040-t001:** Clinical trials (clinicaltrials.gov).

**^1^** **NCT00274846; Donor Peripheral Stem Cell Transplant in Treating Patients With Relapsed Acute Myeloid Leukemia**
**^2^** **NCT00328861; Natural Killer Cells Plus IL-2 Following Chemotherapy to Treat Advanced Melanoma or Kidney Cancer**
**^3^** **NCT01181258; Pentostatin, Rituximab and Ontak and Allogeneic Natural Killer (NK) Cells for Refractory Lymphoid Malignancies**
**^4^** **NCT01385423; Haploidentical Donor Natural Killer Cell Infusion With IL-15 in Acute Myelogenous Leukemia (AML)**
**^5^** **NCT01875601; NK White Blood Cells and Interleukin in Children and Young Adults With Advanced Solid Tumors**
**^6^** **NCT01212341; Allogeneic Natural Killer (NK) Cell Therapy in Patients With Lymphoma or Solid Tumor**
**^7^** **NCT03415100; Pilot Study of NKG2D-Ligand Targeted CAR-NK Cells in Patients With Metastatic Solid Tumours**
**^8^** **NCT03940820; Clinical Research of ROBO1 Specific CAR-NK Cells on Patients With Solid Tumors**
**^9^** **NCT02643550; Study of Monalizumab and Cetuximab in Patients With Recurrent or Metastatic Squamous Cell Carcinoma of the Head and Neck**
**^10^** **NCT03088059; Biomarker-based Study in R/M SCCHN**
**^11^** **NCT02671435; A Study of Durvalumab (MEDI4736) and Monalizumab in Solid Tumors**
**^12^** **NCT02459301; A Dose-Ranging Study of IPH2201 in Patients With Gynecologic Malignancies**
**^13^** **NCT02671435; A Study of Durvalumab (MEDI4736) and Monalizumab in Solid Tumors**
**^14^** **NCT01968109; An Investigational Immuno-therapy Study to Assess the Safety, Tolerability and Effectiveness of Anti-LAG-3 With and Without Anti-PD-1 in the Treatment of Solid Tumors**
**^15^** **NCT03470922; A Study of Relatlimab Plus Nivolumab Versus Nivolumab Alone in Participants With Advanced Melanoma**
**^16^** **NCT02658981; Anti-LAG-3 Alone & in Combination w/ Nivolumab Treating Patients w/ Recurrent GBM (Anti-CD137 Arm Closed 10/16/18)**

NKG2D: Natural Killer Group 2D; CAR: Chimeric Antigen Receptor; ROBO1: roundabout-1; SCCHN: squamous cell carcinoma of the head and neck; LAG: Lymphocyte-activation gene 3.

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
