# Peer review of "Strategies to Augment Natural Killer (NK) Cell Activity against Solid Tumors"

_cancers, 2019, doi:10.3390/cancers11071040_

Round 1
Reviewer 1 Report
In their manuscript, Ziqing Chen et al. review the latest research on strategies to reinforce NK cell antitumor immunity addressing the effects of cytokines, adoptive cell therapy, natural killer cells engagers and NK cell immune checkpoints. While the topic and focus of this reviewer are timely and certainly of interest to a broad readership, its is not clear and at least very arbitrary which “topics” and available data have been selected. The authors should try to balance better on what the focus. In addition, also several other issues must be addressed before this work is eligible for publication:
· The manuscript would largely benefit from including an introduction section. This could briefly introduce the topic per se and general hurdles which need to be overcome to make NK cell therapy in solid cancers more successful (see for example review on „Monoclonal antibody therapy of solid tumors: clinical limitations and novel strategies to enhance treatment efficacy“ by Cruz and Kayser in Biologics 2019). Furthermore, an introduction should also briefly mention the major effector mechanisms of NK cells, and among those the one that so far is most successfully exploited for immunotherapy of cancer, namely the induction of ADCC (see as example “NK-mediated antibody-dependent cell-mediated cytotoxicity in solid tumors: biological evidence and clinical perspectives” by Lo Nigro in Annals of Translational Medicine 2019). In line, the review should be expanded by commenting and summarizing clinically available antitumor antibodies in solid tumors, i.e. Trastuzumab and Cetuximab that mediate their effects, at least in part, by induction of NK cell ADCC.
· The authors should comment on the fact that cytokines like IL-2 and IL-15 lead to NK cell exhaustion when applied as single agents over longer periods of time. They should introduce the concept of immunocytokines in the Cytokine section, and describe that cytokines, immunocytokines and also superagonists like ALT-803 suffer from the problem that they cause relevant side effects that limit applicable doses and thus therapeutic efficacy.
· Besides work that analyzed drug-induced sensitization of tumor cells to NK cell attack (i.e. susceptibility to TRAIL), the authors should summarize the various approaches aiming to increase immunogenicity of tumor cells and thus NK cell recognition, e.g. strategies to induce expression of ligands for activating NK cell receptors by epigenetic therapies, etc..
· Besides describing novel antibody formats like bi- and trispecific antibodies, the authors should comment on Fc-optimization strategies that serve to reinforce NK cell ADCC which presently are increasingly utilized and have entered clinical evaluation.
· The authors should either omit the few sentences on the role of soluble NKG2D ligands or address this complex (differences between mouse and man etc.) and intensively studied field more comprehensively.
· Besides Table 1 which is not referenced in the text, no figures and/or tables are contained. The review would certainly benefit from the inclusion of meaningful figures and/or tables to better summarize the available data/strategies.
· At least in part, the article is written rather negligently, containing a lot of typos, and there are issues with sentence structure and punctuation. The manuscript should be revised accordingly with regard to this matter.
Author Response
Dear Ms. Diana Wang,
thank you for giving us the opportunity to revise our manuscript. We thank the reviewers for their valuable comments and have addressed all comments and concerns below. We have included an introduction, a figure, and the manuscript has been extensively edited with regards to content and language.
Reviewer #1
Including an introduction section. This could briefly introduce the topic per se and general hurdles which need to be overcome to make NK cell therapy in solid cancers more successful. An introduction should also briefly mention the major effector mechanisms of NK cells, and among those the one that so far is most successfully exploited for immunotherapy of cancer, namely the induction of ADCC.
An introduction has now been included in the manuscript to address these concerns.
The review should be expanded by commenting and summarizing clinically available antitumor antibodies in solid tumors, i.e. Trastuzumab and Cetuximab that mediate their effects, at least in part, by induction of NK cell ADCC.
This has now been included on lines 39, 147, and 299.
The authors should comment on the fact that cytokines like IL-2 and IL-15 lead to NK cell exhaustion when applied as single agents over longer periods of time.
This has now been included on lines 65 and 93.
They should introduce the concept of immunocytokines in the Cytokine section, and describe that cytokines, immunocytokines and also superagonists like ALT-803 suffer from the problem that they cause relevant side effects that limit applicable doses and thus therapeutic efficacy.
A section on immunocytokines has now been included on lines 109 through 118. Toxicity related to ALT-803 treatment has been included on lines 105 through 108.
Besides work that analyzed drug-induced sensitization of tumor cells to NK cell attack (i.e. susceptibility to TRAIL), the authors should summarize the various approaches aiming to increase immunogenicity of tumor cells and thus NK cell recognition, e.g. strategies to induce expression of ligands for activating NK cell receptors by epigenetic therapies, etc.
This has now been included on lines 205 through 208.
Besides describing novel antibody formats like bi- and trispecific antibodies, the authors should comment on Fc-optimization strategies that serve to reinforce NK cell ADCC which presently are increasingly utilized and have entered clinical evaluation.
This has now been included on lines 214 through 229.
The authors should either omit the few sentences on the role of soluble NKG2D ligands or address this complex (differences between mouse and man etc.) and intensively studied field more comprehensively.
These sentences have now been omitted.
Besides Table 1 which is not referenced in the text, no figures and/or tables are contained. The review would certainly benefit from the inclusion of meaningful figures and/or tables to better summarize the available data/strategies.
The table has now been appropriately references in the text. A figure describing NK cell immune checkpoints has been included.
At least in part, the article is written rather negligently, containing a lot of typos, and there are issues with sentence structure and punctuation. The manuscript should be revised accordingly with regard to this matter.
The manuscript has been extensively edited with regards to content and language.
Reviewer 2 Report
This is a excellent review on NK-cell based immunotherapy. This comprehensive review covers most of recent important findings in NK cell-based immunotherapy area.
Followings are my suggestions.
This manuscript starts from 1. Cytokine section, but it would be better to start from a brief introduction section (even if there are some overlaps with Abstract).
Including one figure (illustration showing NK-cell checkpoints, NK-based therapeutic strategies) would be very helpful for readers.
3. No need to change the paragraph in Line 243/244(both TIGIT section) and 254/255 (both CIS section).
4. Line177-180 described a controversial point regarding soluble NKG2DL. Antibody-mediated inhibition of MICA/MICB shedding has been proposed to augment ADCC (Science. 2018 Mar 30;359(6383):1537). This information should be included.
Author Response
Dear Ms. Diana Wang,
thank you for giving us the opportunity to revise our manuscript. We thank the reviewers for their valuable comments and have addressed all comments and concerns below. We have included an introduction, a figure, and the manuscript has been extensively edited with regards to content and language.
Rev#2
This manuscript starts from 1. Cytokine section, but it would be better to start from a brief introduction section
An introduction has now been included.
Including one figure (illustration showing NK-cell checkpoints, NK-based therapeutic strategies) would be very helpful for readers.
A figure describing NK cell immune checkpoints has now been included.
Line177-180 described a controversial point regarding soluble NKG2DL. Antibody-mediated inhibition of MICA/MICB shedding has been proposed to augment ADCC (Science. 2018 Mar 30;359(6383):1537). This information should be included.
We agree that this part could be expanded. Based on other reviewers comments we have decided to omit this section. Although the Science. 2018 Mar. paper is an important and groundbreaking paper it does not fall precisely within the scope of this manuscript.
Reviewer 3 Report
This review paper is comprehensive and informative, but is apparently hastily prepared with numerous syntax and stylistic errors. The authors should brush-up the text more carefully. In particular, references are inconsistently listed in the text: they are listed by Roman numbers in parentheses in sections 1 and 2, but parentheses are missing in some cases (for example, line 31), while they are listed by italicized numbers in sections 3 and 4. In some cases a space is missing between a word and a reference number (for example, line 34). Hyphens are often missing when required like single-chain anti-PD-L1 (line 70), antibody-mediated (line 71) and lymphokine-activated killer (line 84). Two consecutive sentences starts with "However," (lines 31 and 33) and administration and administered are repetitious in line 33.
Some sentences do not make sense:
Lines 16-17, what does "immune checkpoint-induced attenuation of antigen presentation" mean? The authors describe that tumors from patients who (should not be "that") progress on anti-PD-1 treatment reveal (should not be "reveals") defects in antigen presentation in lines 13-14. Thus, defects in antigen presentation is no induced by immune checkpoint.
Line 45, what does "superior therapeutic activity to IL-2" mean?
Line 53, "similar immune enhancing properties to IL-2" must be immune enhancing properties similar to IL-2.
Line 59, "anti-tumor activity though a metabolic defect and by limiting IL-15 NK cell function can be restored" does not make sense at all. Do the authors mean anti-tumor activity through a metabolic defect?
Lines 87-88 is not a sentence, and LAK is already defined as an abbreviation in line 84.
Line 99, NK cell immune function or NK cells' immune function.
Line 103, "local or systemic were absent" does not make sense.
Lines 113-115, what starts with "factors regulate" is not a sentence.
Line 121, what does "NK cells were able to resist suppression of MDSCs and Tregs" mean? Suppression by?
Lines 130-132, growth-related oncogene-alpha, beta, and gamma are members of the CXC chemokines.
Lines 140-141, to date and currently are repetitious. What are recruited here?
Line 142, what does "TCRs with specific targeting of melanoma tumors" mean? Targeting to melanoma?
Lines 154-157, the sentence starting with Li et al. doe not make sense at all.
Line 167, Fv has not been defined.
Line 200, KIRs are receptors.
Line 206 "targets that MHC class I-deficient" does not make sense.
Line 237, what is "NKG2A blockade antibody?" Blocking Ab?
Line 240, "termed as damaging the NK cell functions" does not make sense.
Line 244, mouse model, not mice model.
Line 253, CIS is cytokine-inducible SH2ーprotein.
Lines 259-61, IL-1 receptor is IL-1R and IL-1R8 and IL1R8 are inconsistent.
Author Response
Dear Ms. Diana Wang,
thank you for giving us the opportunity to revise our manuscript. We thank the reviewers for their valuable comments and have addressed all comments and concerns below. We have included an introduction, a figure, and the manuscript has been extensively edited with regards to content and language.
Rev#3
We thank this reviewer for a very thorough revision of our manuscript. The manuscript has been extensively edited with regards to content and language and all concerns have been corrected.
The authors should brush-up the text more carefully. In particular, references are inconsistently listed in the text: they are listed by Roman numbers in parentheses in sections 1 and 2, but parentheses are missing in some cases (for example, line 31), while they are listed by italicized numbers in sections 3 and 4.
In some cases a space is missing between a word and a reference number (for example, line 34). Hyphens are often missing when required like single-chain anti-PD-L1 (line 70), antibody-mediated (line 71) and lymphokine-activated killer (line 84). Two consecutive sentences starts with "However," (lines 31 and 33) and administration and administered are repetitious in line 33.
Some sentences do not make sense: Lines 16-17, what does "immune checkpoint-induced attenuation of antigen presentation" mean? The authors describe that tumors from patients who (should not be "that") progress on anti-PD-1 treatment reveal (should not be "reveals") defects in antigen presentation in lines 13-14. Thus, defects in antigen presentation is no induced by immune checkpoint.
Line 45, what does "superior therapeutic activity to IL-2" mean?
Line 53, "similar immune enhancing properties to IL-2" must be immune enhancing properties similar to IL-2.
Line 59, "anti-tumor activity though a metabolic defect and by limiting IL-15 NK cell function can be restored" does not make sense at all. Do the authors mean anti-tumor activity through a metabolic defect?
Lines 87-88 is not a sentence, and LAK is already defined as an abbreviation in line 84.
Line 99, NK cell immune function or NK cells' immune function.
Line 103, "local or systemic were absent" does not make sense.
Lines 113-115, what starts with "factors regulate" is not a sentence.
Line 121, what does "NK cells were able to resist suppression of MDSCs and Tregs" mean? Suppression by?
Lines 130-132, growth-related oncogene-alpha, beta, and gamma are members of the CXC chemokines.
Lines 140-141, to date and currently are repetitious. What are recruited here?
Line 142, what does "TCRs with specific targeting of melanoma tumors" mean? Targeting to melanoma?
Lines 154-157, the sentence starting with Li et al. doe not make sense at all.
Line 167, Fv has not been defined.
Line 200, KIRs are receptors.
Line 206 "targets that MHC class I-deficient" does not make sense.
Line 237, what is "NKG2A blockade antibody?" Blocking Ab?
Line 240, "termed as damaging the NK cell functions" does not make sense.
Line 244, mouse model, not mice model.
Line 253, CIS is cytokine-inducible SH2ーprotein.
Lines 259-61, IL-1 receptor is IL-1R and IL-1R8 and IL1R8 are inconsistent.
Round 2
Reviewer 1 Report
In my opinion the raised concerns were addressed in the revised manuscript
Some minor issues remain:
- Line 11: “can result” as in the former version instead of “results” or another way of attenuation of this statement would be more appropriate
- The authors should carefully revise their use of singular/plural, eg. line 75 “T cells”, line 104 “was/were” etc.
- Sometimes the authors’ writing style is a bit vague sloppy, including but not limited to i) line 91: CD25 region, ii) line 236 wording unclear, iii) do the authors have a source for the Fc-optimized CD33 antibody ? Körner et al 2017 only described a Fc-optimzed CD133 mAb; etc.
After performing these minor amendments, I recommend acceptance
Author Response
- Line 11: “can result” as in the former version instead of “results” or another way of attenuation of this statement would be more appropriate
Has been changed back to "can result".
- The authors should carefully revise their use of singular/plural, eg. line 75 “T cells”, line 104 “was/were” etc.
The manuscript has been thoroughly checked for these errors and been appropriately amended.
- Sometimes the authors’ writing style is a bit vague sloppy, including but not limited to i) line 91: CD25 region, ii) line 236 wording unclear, iii) do the authors have a source for the Fc-optimized CD33 antibody ? Körner et al 2017 only described a Fc-optimzed CD133 mAb; etc.
These errors have been corrected and a reference for the Fc-optimized CD33 antibody has been included.
